# Analysis of TDMP Algorithm of LDPC Codes Based on Density Evolution and Gaussian Approximation

**DOI:** 10.3390/e21050457

**Published:** 2019-05-01

**Authors:** Xiumin Wang, Hong Chang, Jun Li, Weilin Cao, Liang Shan

**Affiliations:** 1College of Information Engineering, China Jiliang University, Hangzhou 310018, China; 2Binjiang College, Nanjing University of Information Science & Technology, Wuxi 214105, China

**Keywords:** density evolution, Gaussian approximation, BP algorithm, TDMP algorithm, convergence speed

## Abstract

Based on density evolution analysis of the existing belief propagation (BP) algorithm, the Turbo Decoding Message Passing (TDMP) algorithm was analyzed from the perspective of density evolution and Gaussian approximation, and the theoretical analysis process of TDMP algorithm was given. When calculating the prior message of each layer of the TDMP algorithm, the check message of the previous iteration should be subtracted. Therefore, the result will not be convergent, if the TDMP algorithm is directly analyzed based on density evolution and Gaussian approximation. We researched the TDMP algorithm based on the symmetry conditions to obtain the convergent result. When using density evolution (DE) and Gaussian approximation to analyze the decoding convergence of the TDMP algorithm, we can provide a theoretical basis for proving the superiority of the algorithm. Then, based on the DE theory, we calculated the probability density function (PDF) of the check-to-variable information of TDMP and its simplified algorithm, and then gave it a calculation based on the process of the normalization factor. Simulation results show that the decoding convergence speed of the TDMP algorithm was faster and the iterations were smaller compared to the BP algorithm under the same conditions.

## 1. Introduction

Low-density parity-check (LDPC) codes were first proposed by Gallager [1] in 1962. In 1996, MacKay and Neal [2] re-researched LDPC codes and discovered that irregular LDPC performs better than regular LDPC [3,4], the irregular LDPC codes can perform close to the Shannon limit error performance. Based on this, Vibha [5] proposed an extended method to convert the regular LDPC codes into the irregular LDPC codes. On the basis of the Slepian–Wolf coding problem, a coding theorem of finite-substitution linear coding was proposed. The research shows that there exist a series of linear coders for any set of finite-correlation discrete memoryless sources [6]. In Reference [7], rate-compatible LDPC coding was implemented by graphics expansion to ensure coding performance and to simplify the structure of the encoder. Based on the probabilistic iterative decoding algorithm, MacKay [8] proposed the belief propagation (BP) algorithm, which is currently of great academic interest. In BP algorithm decoding, information is passed along the edge of Tanner graph until the decoding end [9]. The BP decoding schedules are designed according to the reliability of check nodes, which can improve the error-correcting performance of BP decoding [10]. In addition, only one subset of check nodes is updated in each iteration, which reduces the resource consumption in the decoding process.

Based on Gallager’s [1] thought, Richardson [11] introduced the concept of density evolution, which can calculate the capacity of LDPC codes under message-passing decoding. The density evolution algorithm is also applied to the multi-edge type framework of LDPC [12]. Michael used the density evolution to analyze the convergence of BP iterative decoding and proposed an iterative decoding threshold analysis method for LDPC convolutional codes [13]. In the 5G communication standard [14], Yanming Hao uses the density evolution algorithm to analyze LDPC codes of a sparse-code multiple-access [15] system and designs a multi-user de-noising algorithm. For the binary-input additive white Gaussian noise (BIAWGN) channel, we can replace density evolution (DE) with Gaussian approximation [16] to simplify the analysis of the decoding algorithm. In the performance analysis of low-density parity check-bit interleaved coded modulation system in [17], the probability density function (PDF) of the log–likelihood ratio (LLR) messages transmitted between variables and check nodes is calculated using a Gaussian mixture approximation method for reducing the complexity of the density evolution process. Francesca analyzed the function ϕ(x) involved in Gaussian approximation on the basis in Reference [16] and obtained a new approximation function [18]. The threshold of the new approximation function is closer to that of density evolution compared with the approximation function in Reference [16]. Though the BP algorithm, which is the basic algorithm of LDPC codes, can obtain near Shannon limit error performance, the check nodes need to deal with many non-linear operations, the complexity of the BP algorithm is higher. With further research on the decoding algorithm, Mansour and Shanbhag proposed the Turbo Decoding Message Passing (TDMP) algorithm [19,20], which can obtain high throughput, and at the same time increase the convergence speed of iterative decoding. Under ideal conditions (in an ideal situation), the number of iterations of the TDMP algorithm is half of that of the BP algorithm.

Sachini analyzed the PDF calculation of the Tanner graph edge transfer [21]. Hossein used an Extrinsic Information Transfer Chart (EXIT) to analyze the irregular LDPC codes and proved that there was a similar relationship between regular LDPC and irregular LDPC’s EXIT [22]. In Reference [23], the mixed method of density evolution initialization node and Gaussian approximation calculation could reduce the amount of calculation and facilitate code design. Existing literature on density evolution analysis of LDPC codes are all BP algorithm or derivative algorithms of the BP algorithm, such as the Normalized Min Sum (NMS) algorithm and the Offset Min Sum (OMS) algorithm in References [18,24,25]. Nevertheless, in this paper we present density evolution and Gaussian approximation of the TDMP algorithm based on the BP algorithm, and analyze the decoding convergence of the TDMP algorithm from the perspective of density evolution and Gaussian approximation. Thus, we can provide a more theoretical basis for proving the superiority of the TDMP algorithm. Meanwhile, we simulated six code rates of LDPC codes under the IEEE 802.16 standard and obtained performance curves of the BP algorithm and the TDMP algorithm based on density evolution and Gaussian approximation, respectively. Thereby, we can verify that the convergence speed of the TDMP algorithm was faster. Under the degree distribution of 1/2 code rate, the two algorithms were further compared under the 802.16 and Digital Television Terrestrial Multimedia (DTMB) standards. Then, based on the DE theory, the calculation of the normalized factor is given according to the PDF of the check node information of TDMP and simplified TDMP algorithm, and simulation results are explained.

## 2. The Density Evolution and Gaussian Approximation of the BP Algorithm

### 2.1. The Density Evolution of the BP Algorithm

A regular LDPC code can be represented by (*dv, dc*), where *dv* and *dc* is the degree of variable and check nodes, respectively. The iteration expression of the check messages and the variable messages are as follows [11]
(1)$$\tanh\frac{u_{j}^{\left(l\right)}}{2}=\prod_{i=1}^{dc-1}\tanh\frac{v_{i}^{\left(l-1\right)}}{2},$$
(2)$$v_{i}^{\left(l\right)}=v_{i}^{\left(0\right)}+\sum_{j=1}^{dv-1}u_{j}^{\left(l\right)},$$

$v_{i}^{\left(0\right)}$ represents the initial message of channel received by *i*-th variable node. Assuming that $P^{\left(l\right)}\left(v\right)$ and $Q^{\left(l\right)}\left(u\right)$ are the probability densities of $v_{i}^{\left(l\right)}$ and $u_{j}^{\left(l\right)}$, respectively. We can obtain the result of density evolution from the check messages to the variable messages as [11]
(3)$$P^{\left(l\right)}\left(v\right)=P^{\left(0\right)}⊗Q^{\left(l\right)}{}^{⊗\left(dv-1\right)},$$

⊗ represents the convolution operation. We can transform the convolution operation in Equation (3) to the multiplication operation based on the Fourier transform [11]
(4)$$P^{\left(l\right)}\left(v\right)=F^{-1}\left(F\left(P^{\left(0\right)}\right)\cdot F{\left(Q^{\left(l\right)}\right)}^{dv-1}\right),$$

The process of density evolution from the variable messages to the check messages was introduced by Richardson in detail in Reference [11]. Thus, we can calculate the bit error rate (BER) as
(5)$$P_{e}^{\left(l\right)}=\int_{-\infty }^{0}P^{\left(l\right)}\left(v\right)dv,$$

For irregular LDPC codes, the degree distributions of the variable nodes and the check nodes are $\lambda \left(x\right)=\sum_{i=2}^{dv}\lambda_{i}x^{i-1}$ and $\rho \left(x\right)=\sum_{i=2}^{dc}\rho_{i}x^{i-1}$, respectively. We must consider the degree distribution to update probability density of the variable messages and the check messages [11]. The BER of irregular LDPC codes is also calculated by Equation (5). 

### 2.2. The Gaussian Approximation of the BP Algorithm

For detailed analysis of BP algorithm one can refer to Reference [16]. According to Reference [16], we can get
(6)$$m_{v}^{\left(l\right)}=m_{0}+\left(dv-1\right)m_{u}^{\left(l\right)},$$

$m_{v}^{\left(l\right)}$ and $m_{u}^{\left(l\right)}$ are the means of $v_{i}^{\left(l\right)}$ and $u_{j}^{\left(l\right)}$, respectively, and $m_{0}$ is the mean of channel initial messages. Defining ϕ(x) in Reference [16] and take the expectation of Equation (1) to get Equation (7), where ϕ(x) is shown in Equation (8).
(7)$$m_{u}^{\left(l\right)}=\phi^{-1}[1-{\left(1-\phi (m_{0}+(dv-1)m_{u}^{\left(l-1\right)})\right)}^{dc-1}]$$
(8)$$\phi (x)=\{\begin{array}{c} \sqrt{\frac{\pi }{x}}(1-\frac{10}{7x})e^{-\frac{x}{4}},x\geq 10\\ e^{\left(-0.4527x^{0.86}+0.0218\right)},0<x<10 \end{array}$$

The BER of LDPC codes can be calculated as [21]
(9)$$P_{e}=\int_{-\infty }^{0}\frac{1}{\sqrt{4\pi m_{v}}}\cdot \exp\left(-\frac{{\left(x-m_{v}\right)}^{2}}{4m_{v}}\right)dx.$$

## 3. The Analysis of Density Evolution and Gaussian Approximation of the TDMP Algorithm

### 3.1. The Decoding Procedure of the TDMP Algorithm

Before discussing density evolution and Gaussian approximation of the TDMP algorithm, firstly, we simply review the decoding procedure of the TDMP algorithm [26,27]. Check matrix H with *m* rows and *n* columns is composed of *c* check sub-matrices in series, and each sub-matrix has *k* rows, then m=c×k. Meanwhile, each sub-matrix can be regarded as a horizontal layer, and the main idea of the TDMP algorithm is to decode each layer successively.

In iteration decoding, we first update the check messages in the first layer, and then update the variable messages in this layer immediately. After that, we use the updated variable messages to calculate the prior messages which will be sent to the next layer for updating the check messages until the last layer. Then the updated prior messages are the outputs of iteration decoding. The TDMP algorithm directly uses the updated information of the previous layer when updating the current layer information, thereby speeding up the convergence speed of the decoding. The message passing scheme of the TDMP algorithm is shown in Figure 1.

Decoding the convergence speed can be increased by utilizing the prior messages of the previous layer directly to update the check messages of the current layer. Meanwhile, the essence of layered decoding is optimizing the procedure of message passing but not changing the implementation of the check messages update in each layer. The BP algorithm can also use layered decoding, and the analysis of density evolution and Gaussian approximation of the TDMP algorithm is based on the BP algorithm. Thus, the procedure of the check messages update in each layer of the TDMP algorithm is the same as that of the BP algorithm which is introduced in Section 2. Therefore, in the next section, the major work is to illustrate the variable messages updated in each layer.

### 3.2. The Analysis of Density Evolution of the TDMP Algorithm

The variable messages updated [28,29] in each layer is shown in Equation (10), then prior messages updated in the next layer [30] is shown in Equation (11).
(10)$$v_{j}^{\left(l\right)}=\lambda_{c}+u_{j}^{\left(l\right)},$$
(11)$$\gamma_{j+1}^{\left(l\right)}=v_{j}^{\left(l\right)}-u_{j}^{\left(l-1\right)}=\lambda_{c}+u_{j}^{\left(l\right)}-u_{j}^{\left(l-1\right)},$$

l represents the number of iterations, and j represents the current number of layers. $\lambda_{c}$ is the channel initial message which initializes the variable message before decoding. 

According to the symmetry condition of density evolution [24], we can get Equation (12).
(12)$$\gamma_{j+1}^{\left(l\right)}=\lambda_{c}+u_{j}^{\left(l\right)}+u_{j}^{\left(l-1\right)},$$

When updating to the last layer, the posterior message $\gamma^{\left(l\right)}$ is an output of the *l*-th iteration decoding. 

According to the knowledge related to probability, the probability density of the sum of the two statistically independent variables *X* and *Y* is convolution of probability density of *X* and *Y*. Since the variables in Equations (10) and (12) satisfy the condition of statistically independent distribution, we can obtain
(13)$$P\left(v_{j}^{\left(l\right)}\right)=P\left(\lambda_{c}\right)⊗P\left(u_{j}^{\left(l\right)}\right),$$
(14)$$P\left(\gamma_{j+1}^{\left(l\right)}\right)=P\left(\lambda_{c}\right)⊗P\left(u_{j}^{\left(l\right)}\right)⊗P\left(u_{j}^{\left(l-1\right)}\right),$$

Use the Fourier transform to convert the convolution operation in Equations (13) and (14) into a multiplication operation, as shown in Equations (15) and (16)
(15)$$P\left(v_{j}^{\left(l\right)}\right)=F^{-1}\left(F\left[P\left(\lambda_{c}\right)\right]\cdot F\left[P\left(u_{j}^{\left(l\right)}\right)\right]\right),$$
(16)$$P\left(\gamma_{j+1}^{\left(l\right)}\right)=F^{-1}\left(F\left[P\left(\lambda_{c}\right)\right]\cdot F\left[P\left(u_{j}^{\left(l\right)}\right)\right]\cdot F\left[P\left(u_{j}^{\left(l-1\right)}\right)\right]\right),$$

When updating to the last layer, the probability density $P\left(\gamma^{\left(l\right)}\right)$ of the posterior message $\gamma^{\left(l\right)}$ is the final output probability density at the *l*-th iteration decoding. Thus, we can calculate BER based on Equation (5).

### 3.3. The Analysis of Gaussian Approximation of TDMP Algorithm

According to Section 3.2, we derived Equation (12). By taking expectation of Equations (10) and (12), we can obtain
(17)$$m_{v_{j}}^{\left(l\right)}=m_{0}+m_{u_{j}}^{\left(l\right)},$$
(18)$$m_{\gamma_{j+1}}^{\left(l\right)}=m_{0}+m_{u_{j}}^{\left(l\right)}+m_{u_{j}}^{\left(l-1\right)}.$$

We can calculate BER based on Equation (9).

## 4. Calculation of TDMP Normalization Factor Based on Density Evolution

When calculating the check node information, the TDMP algorithm uses the same nonlinear function as BP, which makes the hardware implementation difficult. The simplified algorithm in Reference [31] simplifies the nonlinear function and provides a possibility for hardware implementation. On the basis of the previous analysis, this paper presents the calculation process of the normalized factor of the simplified TDMP algorithm in Reference [31] based on the density evolution theory. The detailed process is as follows.

### 4.1. The Initialization

Under the BIAWGN channel and Binary Phase Shift Keying (BPSK) modulation, code words $c_{i}$ are mapped to $x_{i}={\left(-1\right)}^{c_{i}}$,i=1,2,⋅⋅⋅n. The received code words are $y_{i}=x_{i}+n_{i}$, $n_{i}$ is independent and identically distributed Gaussian random variable and bilateral power spectral density is $\frac{n_{0}}{2}$. Hence, $y_{i}$ is a Gauss variable with a mean value of 1 and variance of $\sigma^{2}$ [18]. 

Assuming that the average energy of the signal bit transmitted by the channel is Eb, $N_{0}$ is the power spectral density of noise and the code rate is r. The relationship between the signal–noise ratio (SNR) and noise variance is $\sigma^{2}=\frac{1}{2r(Eb/N0)}$.

Suppose that the information is an equal probability distribution [21], where ${Pr(x}_{i}=+1{)=Pr(x}_{i}=-1)=\frac{1}{2}$. The initial information is:(19)$$\begin{array}{l} Pr(c_{i}=1|y_{i})=Pr(x_{i}=-1|y_{i})=\frac{1}{1+e^{2y_{i}/\sigma^{2}}}\\ Pr(c_{i}=0|y_{i})=Pr(x_{i}=+1|y_{i})=\frac{1}{1+e^{-2y_{i}/\sigma^{2}}} \end{array}$$

The log–likelihood ratio message is: (20)$$L(x|y)=\ln(\frac{Pr(x_{i}=+1|y_{i})}{Pr(x_{i}=-1|y_{i})})=\frac{2y}{\sigma^{2}},$$

At this time, the initial information of the LLR is Gaussian random variable [21], whose mean value is $\frac{2}{\sigma^{2}}$ and variance is $\frac{4}{\sigma^{2}}$, that is, it satisfies LLR~N(μ,2μ), where μ=2σ2.

According to the Gaussian distribution of LLR and relevant knowledge of probability theory, its probability density is $f(x)=\frac{1}{\sqrt{4\pi \mu }}\cdot \exp(-\frac{{\left(x-\mu \right)}^{2}}{4\mu })$.

Introduction of the function $Q(x)=\int_{x}^{\infty }q(t)dt$, where $q(t)=\frac{1}{\sqrt{2\pi }}e^{-t^{2}/2}$. Since q(t) is the probability density function of the standard Gaussian distribution, the relationship between q(t) and f(x) is: 12μ⋅q(t)=f(x), where $t=\frac{\left(x-\mu \right)}{\sqrt{2\mu }}$.

### 4.2. Normalization Factor Calculation

A TDMP simplification algorithm without non-linear functions is given in Reference [30], which can be expressed as
(21)$$u=\prod_{j=1}^{dc-1}\mathrm{sgn}(v_{j})\times (\underset{j=1,2\cdots dc-1}{\min}|v_{j}|+r(|v_{j}|)),$$where $r|v_{j}|=\ln(1+e^{-|v_{j}|})\approx \ln2-0.5|v_{j}|$.

The update of the check node information can be simplified to
(22)$$u=\prod_{j=1}^{dc-1}\mathrm{sgn}(v_{j})\times (\underset{j=1,2\cdots dc-1}{\min}|v_{j}|+\ln2-0.5\cdot |v_{j}|),$$

In order to improve the decoding performance of the TDMP simplification algorithm, the normalization factor α is used for improving the calculation of the check node information. Its calculation formula is as shown in Equation (23). The following is the calculation of the normalization factor α based on density evolution [24].
(23)$$u=\prod_{j=1}^{dc-1}\mathrm{sgn}(v_{j})\times \alpha (\underset{j=1,2\cdots dc-1}{\min}|v_{j}|+\ln2-0.5\cdot |v_{j}|),$$
(24)$$\begin{array}{l} L=\prod_{J=1}^{dc-1}\mathrm{sgn}(v_{j})\times (\underset{j=1,2\cdots dc-1}{\min}|v_{j}|+(\ln2-0.5\cdot |v_{j}|))\\ E(L)=E(\underset{j=1,2\cdots dc-1}{\min}|v_{j}|+(\ln2-0.5|v_{j}|))=E(L_{1})+E(L_{2}) \end{array}$$

To facilitate calculation and illustration, variables $L_{1}$ and $L_{2}$ are introduced to represent $\underset{j=1,2,\ldots dc-1}{\min}|v_{j}|$, and $\ln2-0.5|v_{j}|$, respectively.
(25)$$E(L_{1})=E(\underset{j=1,2\cdots dc-1}{\min}|v_{j}|),$$
(26)$$E(L_{2})=E(\ln2-0.5\cdot |v_{j}|),$$
(27)$$E(L_{3})=E(|u|)=2\tanh^{-1}(\prod_{j=1}^{dc-1}\tanh\frac{\left|v_{j}\right|}{2}),$$
(28)$$\alpha =\frac{E(L_{3})}{E(L_{1})+E(L_{2})},$$

According to the knowledge of probability theory, the probability distribution of $\left|L_{1}\right|$ is known [24].
(29)$$Pr(L_{1}>v)=Pr(\min(V_{1},V_{2},\cdot \cdot \cdot ,V_{dc-1})>v)\;=Pr(V_{1}>v,V_{2}>v,\cdot \cdot \cdot ,V_{dc-1}>v)={\left[Pr(V_{1}>v)\right]}^{dc-1},$$

Thus, its expectation is
(30)$$\begin{array}{l} E(L_{1})=\int_{0}^{\infty }Pr(L_{1}>v)dv=\int_{0}^{\infty }{\left[Pr(V_{1}>v)\right]}^{dc-1}dv\\ =\int_{0}^{\infty }{\left[1-F(v)\right]}^{dc-1}dv={\int_{0}^{\infty }\left[1-(1-Q(\frac{\mu +v}{\sigma })-Q(\frac{\mu -v}{\sigma }))\right]}^{dc-1}dv\\ ={\int_{0}^{\mu }\left[1-Q(\frac{\mu -v}{\sigma })+Q(\frac{\mu +v}{\sigma })\right]}^{dc-1}dv+\int_{\mu }^{\infty }{\left[Q(\frac{v-\mu }{\sigma })+Q(\frac{\mu +v}{\sigma })\right]}^{dc-1}dv\\ \approx {\int_{0}^{\mu }\left[1-Q(\frac{\mu -v}{\sigma })+Q(\frac{\mu +v}{\sigma })\right]}^{dc-1}dv \end{array}$$

Since $E(\ln2-0.5|v_{j}|)=\ln2-0.5E(|v_{j}|)$, the PDF is $f(v)=\frac{1}{\sigma \sqrt{2\pi }}e^{-\frac{{\left(v-\mu \right)}^{2}}{2\sigma^{2}}}$,where $\sigma^{2}=2\mu$.

By definition of the mean we have

$E(|v_{j}|)=\int_{-\infty }^{+\infty }|v_{j}|f(v_{j})dv_{j}=\int_{-\infty }^{+\infty }|v_{j}|\frac{1}{\sigma \sqrt{2\pi }}e^{-\frac{{\left(v_{j}-\mu \right)}^{2}}{2\sigma^{2}}}dv_{j}=2{\int_{0}^{+\infty }v}_{j}\frac{1}{\sigma \sqrt{2\pi }}e^{-\frac{{\left(v_{j}-\mu \right)}^{2}}{2\sigma^{2}}}dv_{j}$ which using integral properties can be written as
(31)$$\begin{array}{l} E(|v_{j}|)=2\int_{0}^{+\infty }\left(v_{j}+\mu \right)\frac{1}{\sigma \sqrt{2\pi }}e^{-\frac{v_{j}{}^{2}}{2\sigma^{2}}}dv_{j}=2(\int_{0}^{+\infty }v_{j}\frac{1}{\sigma \sqrt{2\pi }}e^{-\frac{v_{j}{}^{2}}{2\sigma^{2}}}dv_{j}+\mu \int_{0}^{+\infty }\frac{1}{\sigma \sqrt{2\pi }}e^{-\frac{v_{j}{}^{2}}{2\sigma^{2}}}dv_{j})\\ =2(-\sigma \int_{0}^{+\infty }\frac{1}{\sqrt{2\pi }}(-\frac{v_{j}}{\sigma^{2}})e^{-\frac{v_{j}{}^{2}}{2\sigma^{2}}}dv_{j}+\mu \int_{0}^{+\infty }\frac{1}{\sigma \sqrt{2\pi }}e^{-\frac{v_{j}{}^{2}}{2\sigma^{2}}}dv_{j})\\ =2(-\frac{\sigma }{\sqrt{2\pi }}\cdot e^{-\frac{v_{j}{}^{2}}{2\sigma^{2}}}{|}_{0}^{+\infty }+\frac{\mu }{2})=2(\frac{\sigma }{\sqrt{2\pi }}+\frac{\mu }{2})=\mu +\sigma \sqrt{\frac{2}{\pi }} \end{array}$$

And then the calculation of $E(L_{3})$. Extend $\tanh^{-1}$ into a series form
(32)$$\tanh^{-1}(x)=\frac{1}{2}\left[\ln(1+x)-\ln(1-x)\right]=\sum_{k=0}^{\infty }\frac{x^{2k+1}}{2k+1},$$

According to the series expansion [18]
(33)$$E(L_{3})=2E\left[\tanh^{-1}(\prod_{j=1}^{dc-1}\tanh(\frac{\left|v_{j}\right|}{2}))\right]=2\sum_{k=0}^{\infty }\frac{E\left[{\left(\prod_{j=1}^{dc-1}\tanh(\frac{\left|v_{j}\right|}{2})\right)}^{2k+1}\right]}{2k+1}=2\sum_{k=0}^{\infty }\frac{{\left(E\left[{\left(\tanh(\frac{\left|v_{j}\right|}{2})\right)}^{2k+1}\right]\right)}^{dc-1}}{2k+1}$$

According to mathematical knowledge and relevant literature, such as [21,24,25], the high-order expansion terms of the Taylor series are very small (negligible), so the first few terms of the series are usually taken for approximate calculation. In this paper, the first five terms were selected as approximate results by referring to Reference [25]. 

## 5. Simulation Results

The IEEE 802.16e standard defines six code rates (1/2, 2/3A, 2/3B, 3/4A, 3/4B, and 5/6) of the LDPC codes. In this paper, density evolution and Gaussian approximation simulation of LDPC codes with six code rates under 802.16 standard were carried out under the BIAWGN channel. Then, from the perspective of density evolution and Gaussian approximation, the convergence of BP algorithm and TDMP algorithm for 1/2 code rate LDPC codes under 802.16 and DTMB standards was compared. In the simulation, the maximum number of iterations of BP algorithm and TDMP algorithm were set as 20 and 10, respectively. Finally, the calculation of the normalization factor and the performance comparison of the TDMP algorithm were simulated.

### 5.1. The Simulation Results of Density Evolution of the BP Algorithm and the TDMP Algorithm

The performance curves of the BP algorithm and the TDMP algorithm at six code rates are shown in Figure 2a,b respectively. The values of the signal–noise ratio (SNR) of rate 1/2, rate 2/3A (or 2/3B), rate 3/4A (or 3/4B) and rate 5/6 are 0 to 2.5(dB), 0.5 to 3.0(dB), 1.0 to 3.5(dB), and 1.5 to 4.0(dB), respectively.

As can be seen from Figure 2, as the SNR increases, the BER curve of the TDMP algorithm decreased faster than the BP algorithm. The following is a comparison of the convergence of the two algorithms under a single code rate. The code rate 2/3B and SNR 2.0(dB) were chosen in discussion. The relationship between BER and the iterations is shown in Figure 3. Obviously, the BER convergence speed of the TDMP algorithm was faster than the BP algorithm. When the two algorithms achieved the same BER, such as 10^−5^, the number of iterations of the BP algorithm was 13, but the number of the TDMP algorithm was only 7.

When expected BER reaches to 10^−5^ at code rate 2/3B under different SNRs, iterations needed for the BP algorithm and the TDMP algorithm are shown in Figure 4. The BP algorithm and the TDMP algorithm cannot reach to expected BER within the maximum iterations, when SNR are 0, 0.5, 1.0 and 1.5(dB), respectively. When SNR is greater than 1.5(dB), the TDMP algorithm needs fewer iterations compared with the BP algorithm.

From the simulation results of Figure 3 and Figure 4, we can prove that decoding convergence speed of the TDMP algorithm was faster than the BP algorithm from the perspective of density evolution theory.

In order to verify the results, we simulated and analyzed the LDPC codes with 1/2 code rate under the 802.16 standard and the DTMB standard, respectively. The degree distributions of check nodes and variable nodes under the two standards are as follows:

Under the 802.16 standard, the degree distribution of check nodes and variable nodes are shown in Equations (34) and (35)
(34)$$\lambda (x)=0.2895x+0.3158x^{2}+0.3947x^{5}$$
(35)$$\rho (x)=0.6316x^{5}+0.3684x^{6}$$

Under the DTMB standard, the degree distribution of check nodes and variable nodes are shown in Equations (36) and (37)
(36)$$\lambda (x)=0.3382x^{2}+0.2618x^{3}+0.4x^{10}$$
(37)$$\rho (x)=0.1273x^{6}+0.8727x^{7}$$

Figure 5 and Figure 6 compare the BP and TDMP algorithms under the two standards. When the expected BER reaches to 10^−5^, the number of iterations required under different SNRs is shown in Figure 5. When the SNR is 2 dB, the relationship between BER and iteration is shown in Figure 6.

By comparing the simulation results under the two standards in Figure 5 and Figure 6, it can be seen that when the SNR is 2 dB, the convergence speed of TDMP algorithm was faster than that of BP algorithm under the two standards. In addition, as can be seen from Figure 5, when the was greater than 1.5 dB, the number of iterations required by the TDMP algorithm is about 1/2 of the BP algorithm. 

### 5.2. The Simulation Results of the Gaussian Approximation of the BP Algorithm and the TDMP Algorithm

Gaussian approximation algorithm is a simplified density evolution algorithm. Next, the correctness of the TDMP algorithm was verified from the point of view of Gauss approximation. Through Gaussian approximation analysis, the performance curves of the BP algorithm and the TDMP algorithm at six code rates are shown in Figure 7a,b, respectively. It can be seen from Figure 7 that the BER curve of the TDMP algorithm converges faster. When the code rate is 2/3B and the SNR is 2.0 (dB), the relationship between the BER and the number of iterations of the two algorithms is shown in Figure 8. When the expected BER reaches to 10^−5^ at the code rate 2/3B under different SNRs, iterations needed for the BP algorithm and the TDMP algorithm are shown in Figure 9.

The analysis of Figure 7, Figure 8 and Figure 9 is similar as that of Figure 2, Figure 3 and Figure 4 in Section 5.1. Thus, the performance of TDMP algorithm for multi-dimensional density evolution and one-dimensional Gaussian approximation process was similar.

Based on Gaussian approximation theory, the BP and TDMP algorithms of LDPC codes with 1/2 code rate under two standards are simulated. Figure 10 shows the number of iterations required for the two algorithms to achieve the desired BER at different SNRs, where the expected BER is 10^−5^. When the code rate is 1/2 and the SNR is 2.0 dB, the relationship between the BER and the number of iterations is as shown in Figure 11. The standard adopted and the corresponding degree distribution function are the same as Section 5.1.

The analysis of Figure 10 and Figure 11 was similar to that of Figure 5 and Figure 6 in Section 5.1. Combined with the simulation results of Section 5.1 and Section 5.2, we can get the following conclusions. For LDPC codes under different standards, we only need to know the degree distribution of LDPC codes under the current standard. Therefore, for other standard LDPC codes, the density evolution theory can be used to analyze according to the degree distribution function. In terms of LDPC code structure, the TDMP algorithm is applicable to structured layered LDPC codes [32,33]. The currently known standard LDPC codes conform to this feature, so we can conclude that the density evolution and Gaussian approximation of the proposed TDMP algorithm are applicable to different standards.

### 5.3. The Actual Algorithm Simulation of the BP Algorithm and the TDMP Algorithm

In this section, we simulate the LDPC codes with the code length 2304. With the code rates 2/3B and the SNR 3.0 (dB), the relationship between BER and iterations is shown in Figure 12. Here, SNR was not the same as that in Section 5.1 and Section 5.2, because in the actual algorithm simulation, it cannot reach to the expected BER when SNR is 2.0 (dB). Though SNR was different, it did not influence the analysis of the convergence trend of the BP algorithm and the TDMP algorithm. As shown in Figure 12, the BER convergence speed of the TDMP algorithm in the actual algorithm simulation was also faster than the BP algorithm. When the actual algorithm can decode correctly at the code rate 2/3B under different SNRs, the iterations needed for the BP algorithm and the TDMP algorithm are shown in Figure 13.

The analysis of Figure 13 was similar to that of Figure 4 in Section 5.1. The analysis of the results from the actual algorithm simulation can prove the correctness of the density evolution and Gaussian approximation analysis of the TDMP algorithm. Table 1 shows the number of iterations required for the BP and TDMP algorithms in DE, GA, and actual simulation when the code rate was 2/3 under the 802.16 standard. 

The relationship between the number of iterations required by BP and TDMP algorithm can be seen in Table 1 when the desired BER was reached under three simulations. When the SNR was greater than 2 dB, the number of iterations of the BP algorithm was about twice that of the TDMP algorithm. 

### 5.4. The Simulation Results of TDMP Normalization Factor

In this section, we calculated the normalization factor when the SNR was 1.7, 1.8, and 1.9 dB based on density evolution. The simulation results show that the normalization factor was stable at around 0.5, as shown in Figure 14, which can be used in the following simulations. Based on the density evolution process, the relationship between BER and the number of iterations of the BP algorithm, TDMP algorithm, and normalization factor simplification algorithm (TDMP-N) was simulated, the results are shown in Figure 15. Under the AWGN channel, the BER performance of LDPC codes with a code length of 2304 at different SNRs was simulated, as shown in Figure 16. The average number of iterations at different SNRs was simulated as shown in Figure 17. The convergence speed was analyzed from the perspective of density evolution. The simulation results in Figure 15 show that TDMP-N converged slightly faster than TDMP. The simulation results show that the TDMP-N convergence speed was slightly faster than TDMP, and the average iteration number was similar to TDMP. It indicates that the calculation process of TDMP-N was simpler and more convenient for hardware implementation in the case of less loss of BER performance.

## 6. Conclusions

This paper used the density evolution theory and the Gaussian approximation theory to analyze the TDMP algorithm. The TDMP algorithm optimizes information transmission in the decoding process, that is, the update of the current layer information directly uses the updated information of the previous layer, thereby accelerating the convergence speed of the decoding. Based on the density evolution analysis of the existing BP algorithm, this paper analyzed the TDMP algorithm from the perspective of density evolution, giving the theoretical analysis process of the TDMP algorithm, and verifying the analysis results of the TDMP algorithm through simulation comparison with the BP algorithm in this paper. In addition, a simplified TDMP algorithm based on a normalized factor was proposed. The simulation results show that the proposed optimization algorithm had similar performance with the TDMP algorithm. Moreover, it did not involve the operation of non-linear functions and was easier to implement in the hardware. We further simulated six code rates of LDPC codes under the IEEE 802.16e standard and obtained the performance curves of the BP algorithm and the TDMP algorithm based on density evolution and Gaussian approximation, respectively. Thus, we can prove that the decoding convergence speed of the TDMP algorithm was faster than the BP algorithm; meanwhile average iterations were fewer from the perspective of density evolution theory and Gaussian approximation, respectively. According to the structure of LDPC codes under the known standard and the simulation results in the text, we can conclude that the density evolution and Gaussian approximation of the proposed TDMP algorithm can be applied to different standards.

## Figures and Tables

**Figure 1 entropy-21-00457-f001:**
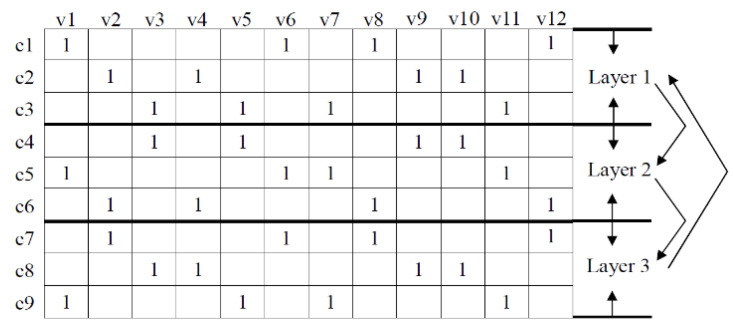
The message passing scheme of the Turbo Decoding Message Passing (TDMP) algorithm.

**Figure 2 entropy-21-00457-f002:**
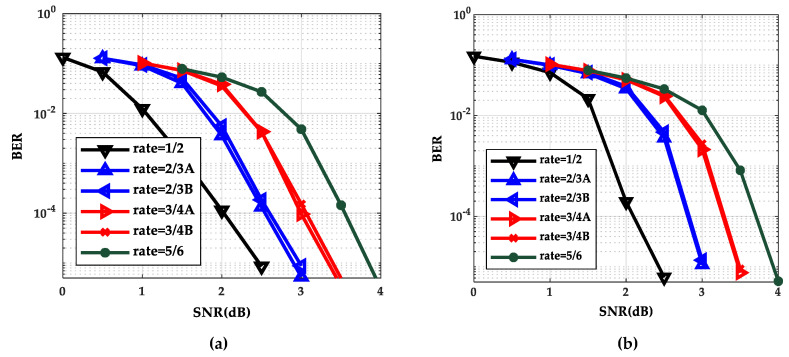
Bit error rate (BER) performance comparison at six code rates of density evolution (DE); (**a**) Belief propagation (BP) algorithm; (**b**) TDMP algorithm.

**Figure 3 entropy-21-00457-f003:**
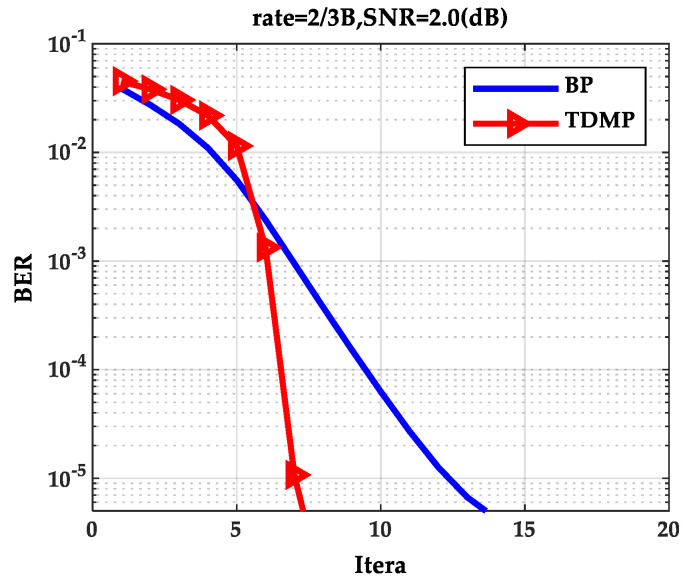
The relationship between BER and iterations.

**Figure 4 entropy-21-00457-f004:**
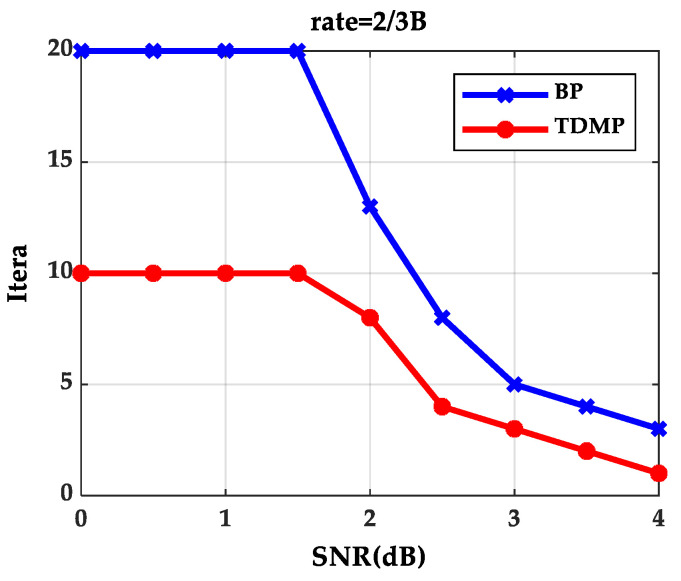
Iterations needed for reaching the expected BER under different signal–noise ratios (SNRs).

**Figure 5 entropy-21-00457-f005:**
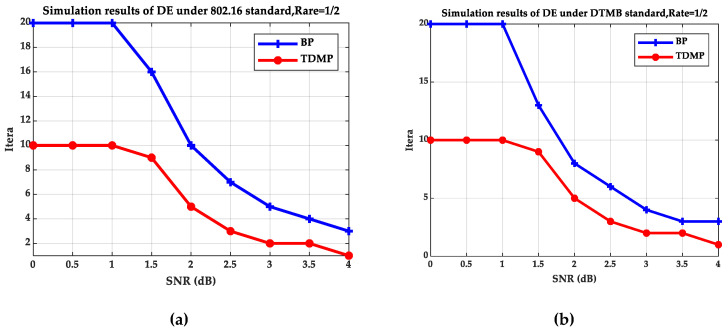
The relationship of SNR and iterations: (**a**) The 802.16 standard; (**b**) the DTMB standard.

**Figure 6 entropy-21-00457-f006:**
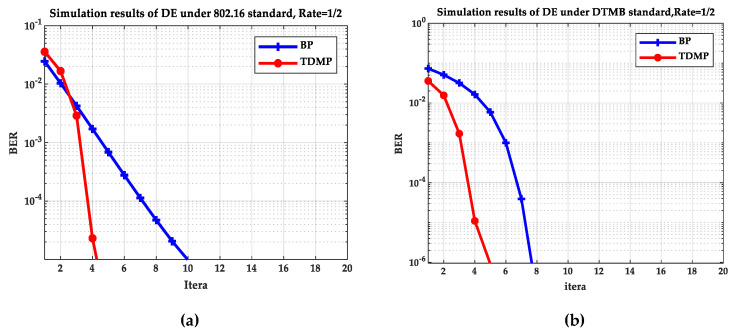
The relationship of BER and iterations: (**a**) the 802.16 standard; (**b**) the DTMB standard.

**Figure 7 entropy-21-00457-f007:**
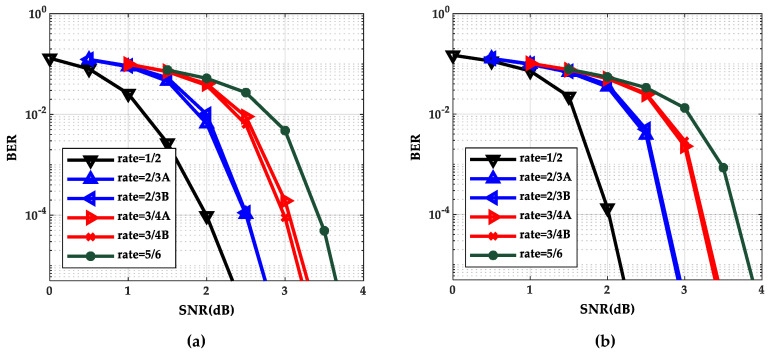
BER performance comparison at six code rates of Gaussian Approximation: (**a**) BP algorithm; (**b**) TDMP algorithm.

**Figure 8 entropy-21-00457-f008:**
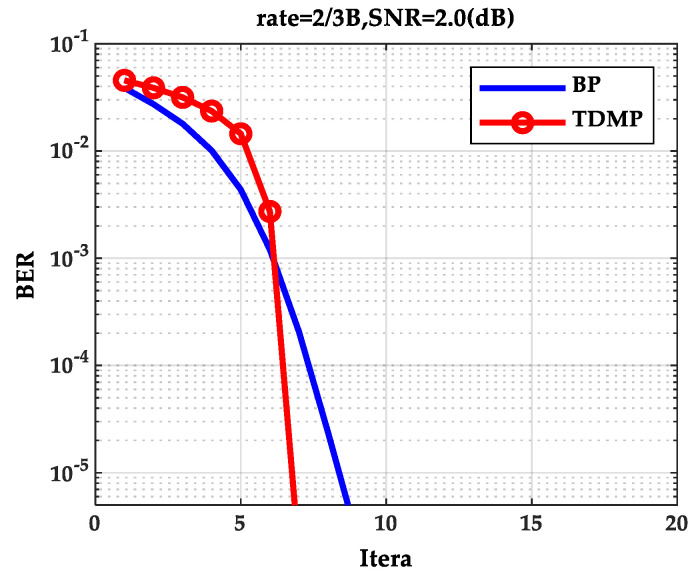
The relationship between BER and iterations.

**Figure 9 entropy-21-00457-f009:**
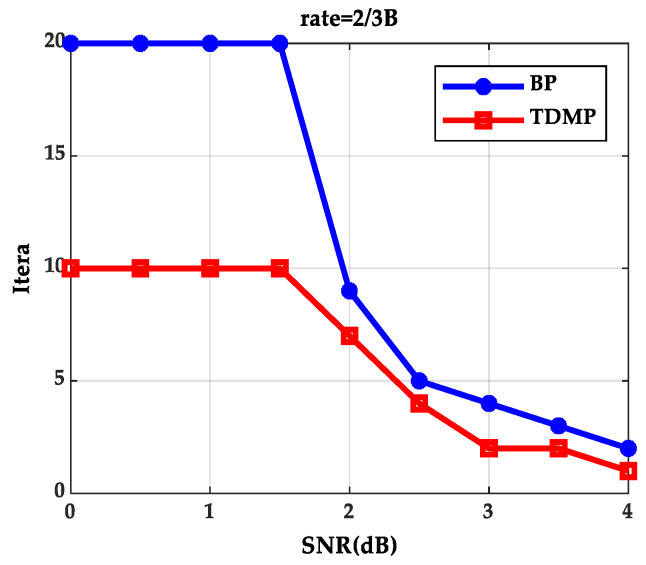
Iterations needed for reaching to the expected BER under different SNRs.

**Figure 10 entropy-21-00457-f010:**
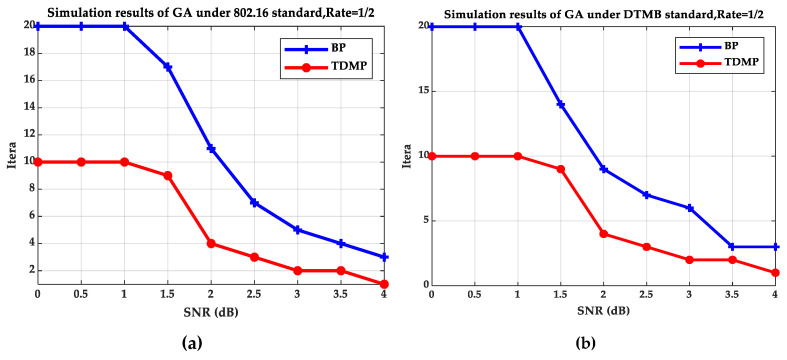
The relationship of SNR and iterations: (**a**) the 802.16 standard; (**b**) the DTMB standard.

**Figure 11 entropy-21-00457-f011:**
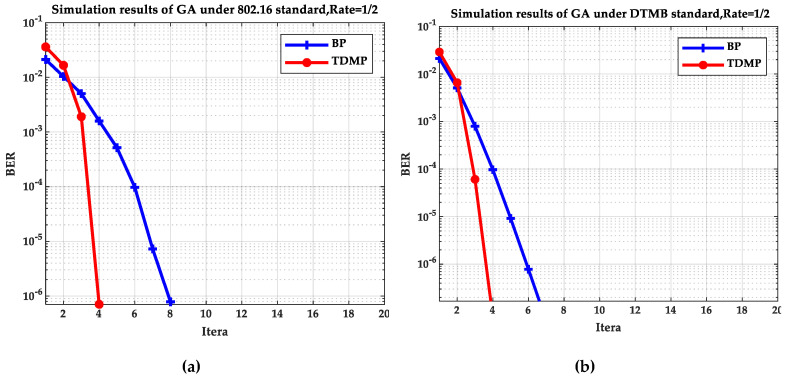
The relationship of BER and iterations: (**a**) the 802.16 standard; (**b**) DTMB standard.

**Figure 12 entropy-21-00457-f012:**
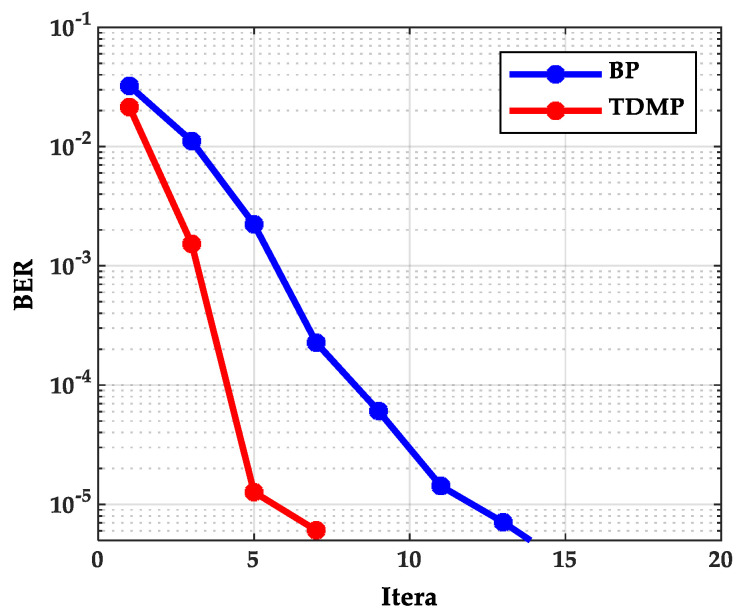
The relationship between BER and iterations.

**Figure 13 entropy-21-00457-f013:**
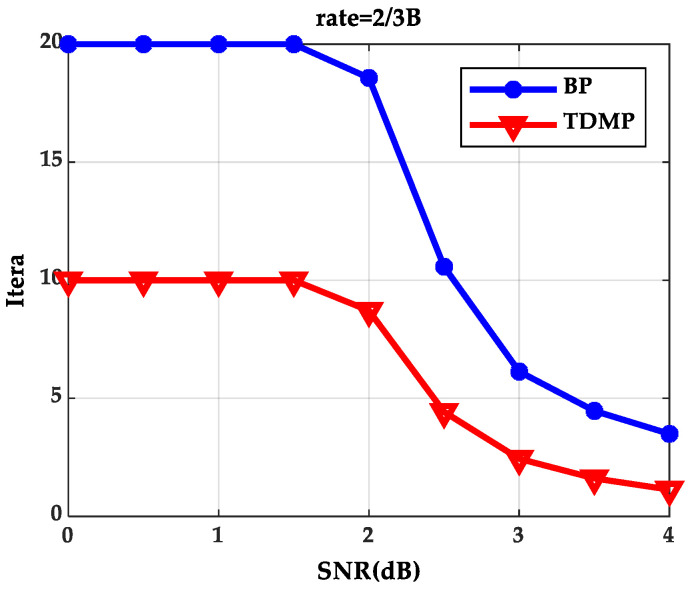
Iterations needed for decoding correctly under different SNRs.

**Figure 14 entropy-21-00457-f014:**
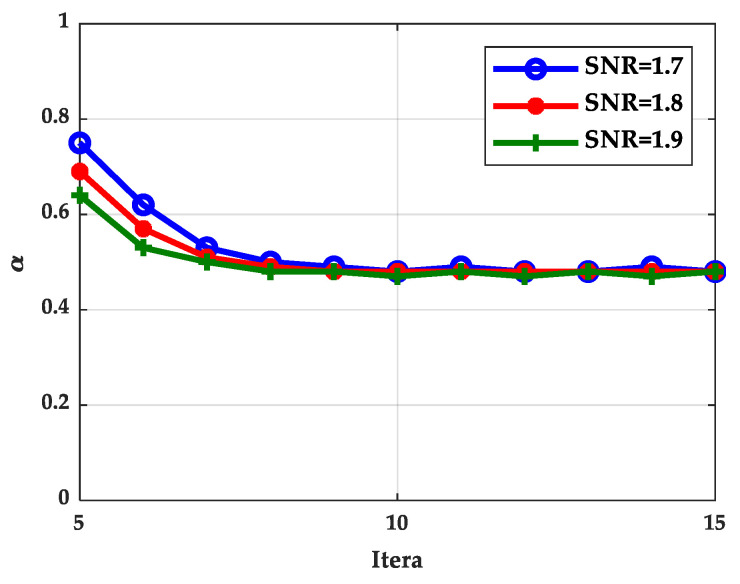
The normalized factor calculated under different iteration times.

**Figure 15 entropy-21-00457-f015:**
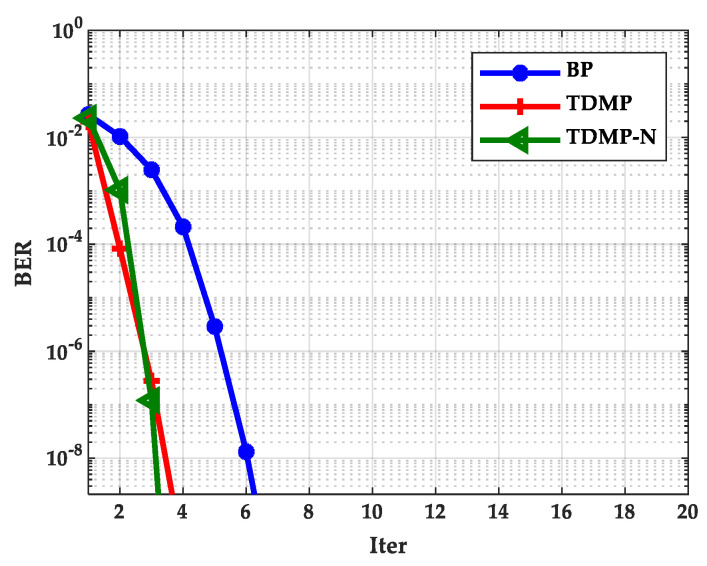
The relationship between different iteration times and BER.

**Figure 16 entropy-21-00457-f016:**
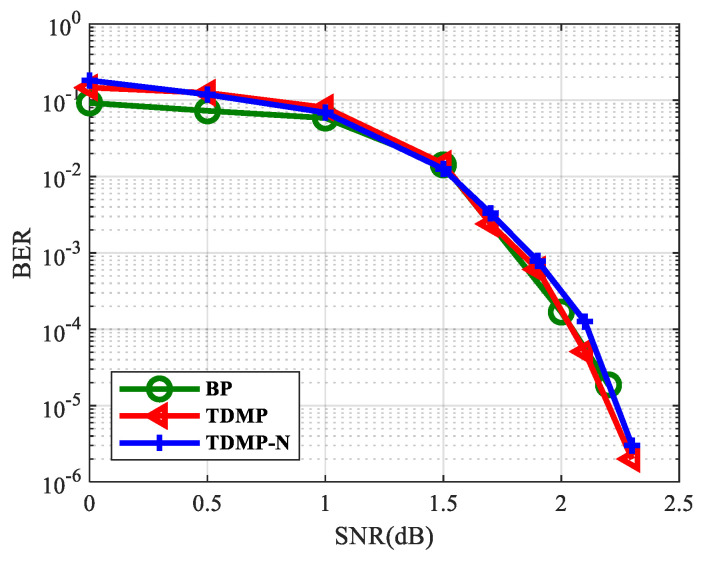
The relationship between BER and SNR.

**Figure 17 entropy-21-00457-f017:**
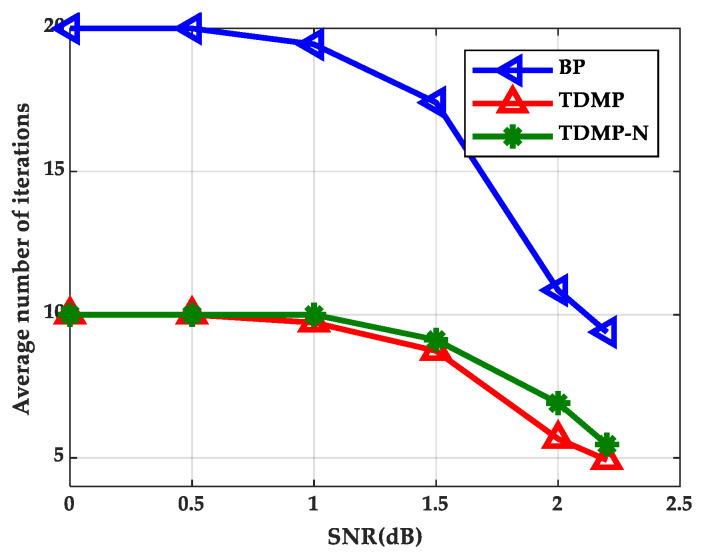
The relationship between the average number of iterations and SNR.

**Table 1 entropy-21-00457-t001:** Iteration times of BP and TDMP algorithms under different SNRs in DE, GA and actual simulation.

Iteration SNR (dB)		0 dB	1 dB	2 dB	3 dB	4 dB
DE	BP	20	20	13	5	3
	TDMP	10	10	7	3	1
GA	BP	20	20	9	4	2
	TDMP	10	10	7	2	1
Actual	BP	20	20	18	6	4
	TDMP	10	10	8	3	2

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
