# Peer review of "Analysis of TDMP Algorithm of LDPC Codes Based on Density Evolution and Gaussian Approximation"

_entropy, 2019, doi:10.3390/e21050457_

Round 1
Reviewer 1 Report
Authors presented the final theoretical justification of decoding algorithm based on Gaussian approximation.The calculation process of the normalization factor is also presented .But restriction by five terns of the series expansian is not explained.Simulation results confirms theoretical investigation and the fact more faster decoding convergence of TDMP algorithm against BP algorithm.Abstract can be shorted and part of it moved to Introduction.Some English phrases should be improved or corrected .So , on the page 7, the second row , just after Fig. 2 it is necessary to insert at the end of this row , the words: -"the number of".
Author Response
Response to Reviewer 1 Comments
Dear honorable editors and professors:
Thank you for giving us helpful comments and advice about this paper. We have already read the comments and revised the paper carefully. We edit the paper according to the submission template to standardize the article format. All the modifications were marked in blue in our new manuscript.
Point 1: But restriction by five terns of the series expansion is not explained.
Response 1: Thank you for this comment. In [21], [29] and [31], it is pointed out that for the infinite expansion of Taylor series, the value of the higher-order expansion is small (insignificant) [21], and the sum of the first several terms is generally taken as the result. Therefore, the first five terms are taken as the result like the reference [31].
[21] Sachini Jayasooriya; Mahyar Shirvanimoghaddam; Lawrence Ong; Gottfried Lechner; Sarah J. Johnson. A New Density Evolution Approximation for LDPC and Multi-Edge Type LDPC Codes. IEEE Trans. Commun. 2016, 64, 4044 – 4056
[29] Xiumin Wang; Weilin Cao; Jun Li; Liang Shan; Haiyan Cao; Jinsong Li; Fanglei Qian. Improved min-sum algorithm based on density evolution for low-density parity check codes. IET Commun. 2017, 11, 1582-1586
[31] Jinghu Chen, Marc P. C. Near Optimum Universal Belief Propagation Based Decoding of Low-Density Parity Check Codes. IEEE Trans. Commun 2002, 50(3): 406-414
Point 2: Abstract can be shorted and part of it moved to Introduction.
Response 2: Thank you for this comment and we have revised abstract.
Point 3: So, on the page 7, the second row, just after Fig. 2 it is necessary to insert at the end of this row, the words: "the number of".
Response 3: Thanks for your comment, we have made improvement according to your opinion.
We have changed “When the two algorithms achieve the same BER, such as 10-5, the number of iterations of BP algorithm is 13, but that of the TDMP algorithm is only 7.” into “When the two algorithms achieve the same BER, such as 10-5, the number of iterations of BP algorithm is 13, but the number of the TDMP algorithm is only 7.”

Reviewer 2 Report
Review
In this manuscript, the author proposes an improved TDMP algorithm based on the BP algorithm using density evolution and Gaussian approximation. Theoretical basis is provided by analyzing the convergence of the TDMP algorithm. The algorithm involving no non-linear operation is faster and easier to implement in hardware. The limitations of the manuscript are as follows:
1) Based on the title or introduction part, I cannot grasp the main point of the manuscript the authors would like to address. Specifically, what is the relation between BP and TDMP algorithm? Since that in the experiments, two algorithms are extensively compared , while not with current arts. Why?
2) In the introduction, the authors just present some algorithms while the logical expression cannot be given. What are the limitations of current arts? Why do the authors propose the study of an improved TDMP algorithm?
In the following paragraphs, I will specifically extend my comments.
Major Comments:
1) The title "Analysis of TDMP Algorithm of LDPC Codes Based on Density Evolution and Gaussian Approximation", do authors of this manuscript just make analysis of the TDMP algorithm? If not, the authors may re-organize the title so that the title reflects directly the authors’ contributions.
2) In Page 1Line 9 and Page 11Line308,"In this paper, we present……In this paper, we have presented……", those two sentences are the first sentence of the Abstract and the Conclusion thus they are fundamental and significant to the manuscript. Please re-write those two sentences sharply to emphasize novelties of this paper. By the way, the density evolution and Gaussian approximation are not presented by the authors of this manuscript while the authors just get inspired by them.
3) In Figures14-17, the curves in one Figure could be drawn in different colors.
4) In Page 12 Line 320, "Moreover, density evolution and Gaussian approximation of the TDMP algorithm presented in this paper are also suitable for other standards", which part in this manuscript proves it? In section 5, experiments are only conducted over 802.16 standard and DTMB standard.
In my humble opinion, this manuscript is not qualified to publish on the this journal.
Author Response
Response to Reviewer 2 Comments
Dear honorable editors and professors:
Thank you for giving us helpful comments and advice about this paper. We have already read the comments and revised the paper carefully. We edit the paper according to the submission template to standardize the article format. All the modifications were marked in blue in our new manuscript.
Point 1: The title "Analysis of TDMP Algorithm of LDPC Codes Based on Density Evolution and Gaussian Approximation", do authors of this manuscript just make analysis of the TDMP algorithm? If not, the authors may re-organize the title so that the title reflects directly the authors’ contributions.
Response 1: Thank you for this comment. Existing literatures on density evolution (DE) analysis of LDPC codes are all BP algorithm or derivative algorithms of BP algorithm, such as NMS algorithm and OMS algorithm, such as literature [18], literature [29], literature [31], etc., while analysis of TDMP algorithm based on DE is rarely analyzed.
This paper analyzes the TDMP algorithm from the perspective of density evolution on the basis of analysis of the existing BP algorithm, gives the theoretical analysis process of TDMP algorithm, and verifies the analysis result of this paper through simulation comparison with BP algorithm. The TDMP algorithm optimizes the way of information transmission in the decoding process, that is, the update of the current layer information directly uses the updated information of the previous layer, thereby accelerating the convergence speed of the decoding.
Therefore, we propose that the probability calculation of variable nodes in the current layer is related to the previous layer, and give its calculation formula as in Equations 12 and 14.
Based on the information transmission process of TDMP algorithm in the decoding process, we first carried out density evolution simulation to compare the convergence speed of BP algorithm and TDMP algorithm. The simulation results of density evolution show that the convergence speed of TDMP algorithm is greater than that of BP algorithm. Subsequently, the Gaussian approximation simulation of the TDMP algorithm and the BP algorithm is carried out. It can be seen from the Gaussian approximation simulation results that the convergence speeds of the two algorithms are consistent under the density evolution and Gaussian approximation. Combined with the actual simulation results, the correctness of the density evolution and Gaussian approximation analysis of the proposed TDMP algorithm in this paper is proved.
[18] Francesca Vatta; Alessandro Soranzo; Fulvio Babich. More Accurate Analysis of Sum-Product Decoding of LDPC codes Using a Gaussian Approximation. IEEE Commun. Lett. (Early Access)2018, 1-4
[29] Xiumin Wang; Weilin Cao; Jun Li; Liang Shan; Haiyan Cao; Jinsong Li; Fanglei Qian. Improved min-sum algorithm based on density evolution for low-density parity check codes. IET Commun. 2017, 11, 1582-1586
[31] Jinghu Chen, Marc P. C. Near Optimum Universal Belief Propagation Based Decoding of Low-Density Parity Check Codes. IEEE Trans. Commun 2002, 50(3), 406-414
Point 2: In Page 1Line 9 and Page 11Line308,"In this paper, we present……In this paper, we have presented……", those two sentences are the first sentence of the Abstract and the Conclusion thus they are fundamental and significant to the manuscript. Please re-write those two sentences sharply to emphasize novelties of this paper.
Response 2: Thank you for your comment. We think what you said is quite reasonable, and modify it as follows:
In abstract, we have changed “In this paper, we present density evolution and Gaussian approximation of the TDMP algorithm based on the BP algorithm” into “Based on density evolution analysis of existing BP algorithm, TDMP algorithm is analyzed from the perspective of density evolution and Gaussian approximation, and the theoretical analysis process of TDMP algorithm is given”.
In conclusion, we have changed “In this paper, we have presented density evolution and Gaussian approximation of the TDMP algorithm based on the BP algorithm” into “This paper uses the density evolution theory and the Gaussian approximation theory to analyze the TDMP algorithm. The TDMP algorithm optimizes the way of information transmission in the decoding process, that is, the update of the current layer information directly uses the updated information of the previous layer, thereby accelerating the convergence speed of the decoding. Based on the density evolution analysis of the existing BP algorithm, this paper analyzes the TDMP algorithm from the perspective of density evolution, gives the theoretical analysis process of TDMP algorithm, and verifies the analysis result of TDMP algorithm through the simulation comparison with BP algorithm in this paper.”
Point 3: By the way, the density evolution and Gaussian approximation are not presented by the authors of this manuscript while the authors just get inspired by them.
Response 3: Thank you for this comment. This paper uses the density evolution theory and the Gaussian approximation theory to analyze the TDMP algorithm. Existing literature on density evolution analysis of LDPC codes are all BP algorithm or derivative algorithms of BP algorithm, such as NMS algorithm and OMS algorithm, such as literature [18], literature [29], literature [31], etc., while TDMP algorithm based on DE is rarely analyzed. At present, the convergence analysis of TDMP algorithm is limited to actual simulation. This paper gives a theoretical explanation of TDMP algorithm, which give the probability calculation of variable nodes in the current layer.
Compared with the existing BP algorithm, analysis and relevant simulations of TDMP algorithm based on DE theory are given. The TDMP algorithm optimizes the way of information transmission in the decoding process, that is, the update of the current layer information directly uses the updated information of the previous layer, thereby accelerating the convergence speed of the decoding. Therefore, the probability of the current variable node information is related to the previous layer. This paper proposes to process the probability density of the variable node as shown in Equations 12 and 14 at the section 3.2 in the text.
[18] Francesca Vatta; Alessandro Soranzo; Fulvio Babich. More Accurate Analysis of Sum-Product Decoding of LDPC codes Using a Gaussian Approximation. IEEE Commun. Lett. (Early Access)2018, 1-4
[29] Xiumin Wang; Weilin Cao; Jun Li; Liang Shan; Haiyan Cao; Jinsong Li; Fanglei Qian. Improved min-sum algorithm based on density evolution for low-density parity check codes. IET Commun. 2017, 11, 1582-1586
[31] Jinghu Chen, Marc P. C. Near Optimum Universal Belief Propagation Based Decoding of Low-Density Parity Check Codes. IEEE Trans. Commun 2002, 50(3), 406-414
Point 4: In Figures14-17, the curves in one Figure could be drawn in different colors.
Response 4: Thanks for your comment. Following the suggestion of the referees, we have redrawn Figures 14-17.
Point 5: In Page 12 Line 320, "Moreover, density evolution and Gaussian approximation of the TDMP algorithm presented in this paper are also suitable for other standards", which part in this manuscript proves it? In section 5, experiments are only conducted over 802.16 standard and DTMB standard.
Response 5: Thank you for this comment. When performing density evolution analysis, it is only necessary to know the degree distribution function of LDPC codes, and does not need a specific check matrix, such as Equations 3-5 and 11-14 in the text. The difference between LDPC codes under different standards is only the difference in the degree distribution. The mechanism of information transmission between the variable nodes and the check nodes does not change during the decoding process. Therefore, for LDPC codes under different standards, we only need to know the degree distribution of LDPC codes under current standard. Therefore, for other standards of LDPC codes, the density evolution theory can be used to analyze according to the degree distribution function. In terms of LDPC codes structure, TDMP algorithm is suitable for structured layered LDPC codes [32][33]. Currently known standards for LDPC codes all conform to this feature, so TDMP algorithm can be adopted.
[32] Shuangqu HUANG; Xiaoyang ZENG; Yun CHEN. A Flexible LDPC Decoder Architecture Supporting TPMP and TDMP Decoding Algorithms. IEICE TRANS. ON INF.SYST, 2012, E95D(2): 403-412
[33] Shixian Li; Qichen Zhang; Yun Chen; Xiaoyang Zeng. A High-Throughput QC-LDPC Decoder for Near-Earth Application[C]. 2018 IEEE 23rd International Conference on Digital Signal Processing (DSP), Shanghai, China, Nov. 2018

Reviewer 3 Report
This paper presents two contributions to the decoding of LDPC codes. The first one is theoretical and serves to give additional evidences of the superiority of TDMP decoding with respect to BP decoding. A new analysis is presented of the convergence of TDMP based on density evolution. The Gaussian approximation is also considered. The second contribution is more practical, a new form of computing the normalization factor alleviates the implementation of the TDMP algorithm.
The paper is generally well written, the theoretical analysis seems to be correct and the number and type of experiments are appropriate to validate the claims of the authors. The contributions are reasonably novel and interesting in the area of application. Therefore I consider that the manuscript can be accepted, subject to some minor revisions:
-Do not define acronyms in the abstract that will not be used in the own abstract
-In the last paragraph of the introduction, it should also be mentioned the new contribution in Section 4 regarding the normalization factor.
-Equation notation must be carefully revised. In general the symbols are too small, in some cases misaligned with the text, italic font is not always maintained (for example eq. (32)-(35)), and so on…
Author Response
Response to Reviewer 3 Comments
Dear honorable editors and professors:
Thank you for giving us helpful comments and advice about this paper. We have already read the comments and revised the paper carefully. We edit the paper according to the submission template to standardize the article format. All the modifications were marked in blue in our new manuscript.
Point 1: Do not define acronyms in the abstract that will not be used in the own abstract.
Response 1: Thank you for this comment and we have deleted.
Point 2: In the last paragraph of the introduction, it should also be mentioned the new contribution in Section 4 regarding the normalization factor.
Response 2: Thanks for your comment, we have made improvement according to your opinion.
we have added “Then, based on the DE theory, the calculation of the normalized factor is given according to the PDF of the check node information of the TDMP algorithm and the simplified algorithm of TDMP, and the simulation result is explained” to the introduction
Point 3: Equation notation must be carefully revised. In general the symbols are too small, in some cases misaligned with the text, italic font is not always maintained (for example eq. (32)-(35)), and so on…
Response 3: Thank you for your comments. We agree with your comment very much and we have changed.

Round 2
Reviewer 2 Report
I have no further comments.